# Effects of Biochar and Apatite on Chemical Forms of Lead and Zinc in Multi-Metal-Contaminated Soil after Incubation: A Comparison of Peanut Shell and Corn Cob Biochar

**Truong Xuan Vuong [1],\***, **Thi Thu Ha Pham [1]**, **Thi Thu Thuy Nguyen [1]** and **Dung Thuy Nguyen Pham [2,3],\***

1    Faculty of Chemistry, TNU-University of Science, Tan Thinh Ward, Thai Nguyen City 24000, Vietnam; haptt@tnus.edu.vn (T.T.H.P.); thuyntt@tnus.edu.vn (T.T.T.N.)

2    NTT Institute of Applied Technology and Sustainable Development, Nguyen Tat Thanh University, Ho Chi Minh City 70000, Vietnam

3    Faculty of Environmental and Food Engineering, Nguyen Tat Thanh University, Ho Chi Minh City 70000, Vietnam

\*    Correspondence: xuanvt@tnus.edu.vn (T.X.V.); pntdung@ntt.edu.vn (D.T.N.P.)

**Abstract:** Heavy metal pollution in soils caused by mining activities is a severe issue worldwide. It is necessary to find a suitable approach to mitigate heavy metal-contaminated soil. Yet little is known about how soil amendments affect the chemical forms of heavy metals. Biochar produced from peanut shells (PSB300) and corn cob (CCB300) at 300 °C, and apatite (AP) were applied at various ratios to investigate their ability to adsorb lead (Pb) and zinc (Zn) in contaminated soil. The Pb and Zn's chemical fractions were analyzed utilizing Tessier's sequential extraction procedure and quantified using inductively coupled plasma mass spectroscopy. The one-month amendment incubation of biochar and AP could significantly diminish Pb and Zn's exchangeable fractions, and CCB300 showed a slightly better effect on declining the exchangeable fractions of Pb and Zn than PSB300, which might be attributed to the higher values of OC and EC of CCB300 than those of PSB300. Moreover, the amendments could also transform the exchangeable fractions of Pb and Zn into stable fractions, resulting in immobility in natural conditions. Thus, PSB300 and CCB300 and the mixture of biochar/apatite could be hopeful amendments for immobilizing heavy metals in heavy metal multi-metal-contaminated field soil.

**Keywords:** heavy metal; remediation; chemical fractionation; adsorption; soil amendment

## 1. Introduction

Heavy metal contamination is a serious global problem. A serious environmental problem is the accumulation of heavy metals in the soil brought on by industrial and human activities [1,2]. The main reasons for heavy metal pollution in soils have been attributed to various factors, such as mining and smelting operations or the use of pesticides and sewage sludge [3–5]. Heavy metals can accumulate and remain in the soil for an extended period since they are not biodegradable [5]. Moreover, they can be mobile in soils depending on their chemical state in the soil and different physicochemical processes [6]. As a result, they can intrude into the food chain and pose a grave risk to human health because of their detrimental impacts [4,7–9]. The two most prevalent heavy metal pollutants in agricultural soils are lead (Pb) and zinc (Zn). Lead and zinc are frequently found together in nature, particularly in paddy soils near lead and zinc mines [10].

Several methods have been applied to remediate heavy metal pollution in contaminated soil, such as physical, chemical, biological, and phytoremediation [5,11]. Biochar has been employed in both experimental and field scales to remediate heavy metals (HMs) in contaminated soils, which is considered an economical and environmentally beneficial rehabilitation method [12]. Biochar is produced when organic materials are thermally

transformed in an atmosphere lacking oxygen [13]. Due to biochar's variety of intrinsic properties, its use as a sorbent material has gained more potential for investigation among environmentalists [14]. Biochar is one of the most effective materials for handling and removing various heavy metals due to its properties, including large and porous surface area, high pH, surface functional groups, and ash and carbon content [9,15]. Temperature, pyrolysis time, the feedstock used to make the biochar, and the ultimate acidity significantly impact the resulting material's characteristics [15,16]. Various biochar types have been pyrolyzed at temperatures ranging from about 350 °C to over 750 °C when made from feedstocks like woody residues, agricultural wastes, animal manures, sewerage sludge, and food wastes [13,16,17]. Previous studies have reported the diverse impacts of various biochars on heavy metal remediation in contaminated soil and the efficiency depended on many factors such as heavy metal properties, soil types, application ratios, incubation, types of biomass, pyrolysis temperatures, and time [13,18]; however, the possible mechanism for immobilizing heavy metals in soil using biochar is still an open question since it is an intrinsic process [15], even though many previous studies have informed that the primary mechanisms of the immobilizing process consist of electrostatic attraction, ion exchanges, adsorption, complexation, reduction, and precipitation reactions [9,12,17,19].

Combining biochar with phosphorous-rich minerals such as apatite ore (AP) has been a promising approach to enhancing the immobilizing ability of biochar by adding more inorganic elements, including the phosphate group. AP is frequently used to make phosphate fertilizer and its surface has a negative charge that allows cationic heavy metals to be adsorbed, primarily via ion exchange [20]. Several previous studies reported positive results when combining biochar derived from rice straw produced at 350–500 °C with apatite ore in immobilizing Pb and Zn in tailing soil [20,21]; modifying biochar derived from rape straw produced at 500 °C with phosphorous compounds in remediating Pb, Cd, and Cu in contaminated soil [22]; or combining biochar made from peanut shell at 400 °C and 600 °C with apatite or blending biochar-derived corn cob created at 400 °C and 600 °C with apatite in remediating Pb and Zn in contaminated soil [23,24]. Through a variety of processes, such as direct metal(loid) adsorption or substitution by P compounds, P anion-induced metal(loid) adsorption, and precipitation of metal(-loid)s with solution P as metal(loid) phosphates, phosphate compounds improve the immobilization of heavy metals in soils [25]. Since the effects of biochar in remediating heavy metals depend on various factors, including soil types, heavy metals, biomass, application rates, and pyrolysis temperature [13,18], it is worth noting that different biochars have different physicochemical properties, thus showing diverse effects on the chemical fractions of heavy metals [26]. For example, biochar produced at 300 °C had a higher concentration of soluble phosphate than 400 °C pyrolyzed biochar. As a result, it was more effective at adsorbing lead in the exchangeable fraction than 400 °C pyrolyzed biochar [27]. The number of studies investigating the effects of the blend of biochar and apatite on the chemical fractions of heavy metals in soil in the aspect of various biomass, pyrolysis temperature, and soil types has been limited so far. Therefore, it is essential to focus on combining biochar with AP in different effects of various biomass, pyrolysis temperature, and soil types.

Adding the various amendments did not modify the polluted soil's primary chemical characteristics except for a few things (pH, electrical conductivity, and exchangeable Na and K). Therefore, soil amendments caused pollutants to be redistributed from the solution phase to the solid phase, reducing their bioavailability and transit in the environment [25]. Thus, it is necessary to scrutinize the chemical fractions of HMs in the incubated soil after incubation with biochar to study the efficiency of amendments in remediating heavy metals in contaminated soil. Numerous earlier studies have claimed that sequential extraction is an essential technique that may offer helpful information on the bioavailability of metals and their transition forms [15]. In many previous studies, Tessier's sequential extraction method has been applied to study the chemical speciation of heavy metals in soil [20,21,24,25,28]. According to Tessier, geochemical forms of soil/sediment include the five primary fractions, which are exchangeable, bound to carbonates, bound to iron and manganese oxides, bound to

organic matter, and residual fractions [29] (Tessier 1979). For exchangeable form, metals in this form bind to colloidal particles in soil/sediments (clay, hydrate of iron oxide, manganese oxide, and humic acid) by weak adsorption force. Changes in the ionic strength of water will affect the adsorption or desorption of these metals, leading to metal release or accumulation at the contact surface of water and sediment. Therefore, metals in sediments in this form are very mobile and can be quickly released back into the aquatic environment. For bound to carbonates: metals exist in the form of carbonate salt precipitates. Metals existing in this form are susceptible to changes in pH. When the pH decreases, metals in this form will be released. For bound to an iron and manganese oxide fraction: in this fraction, the metal is adsorbed on the surface of Fe-Mn oxide hydroxide and is unstable under reduced conditions because, under these conditions, the oxidation state of iron and manganese will be altered, resulting in metals in the soil/sediment being released into the aqueous phase. For bound to organic: metals in the form of organic bonds will be unstable under oxidizing conditions. When oxidized, organic substances decompose, and metals are discharged into the aqueous phase. For the residual fraction: this part contains naturally occurring mineral salts that can keep traces of metals in their structural background. As a result, when metals exist in this fraction, they will not be dissolved in water under natural conditions. Therefore, investigating the chemical forms of heavy metals in contaminated soil using Tessier's sequential extraction procedure is essential when ascertaining the effect of biochar amendments on the chemical speciation of heavy metals.

The Hich Pb/Zn mine in Dong Hy district, in Thai Nguyen City, Vietnam, has been reported as a highly polluted heavy metals area caused by the mining activity, especially for lead (Pb) and zinc (Zn), where their concentrations have been reported to range from 2000 to 4000 mg kg$^{-1}$, being about 43 and 10 times greater than the permissible limits set up by the Vietnamese Regulation [23,28]. Hence, it is urgent to achieve a suitable solution to remediate heavy metals in the soil of this area, particularly in agricultural soil. Previous studies investigated the effects of biochar produced from peanut shells and corn cob at 400 °C and 600 °C and their combination with AP on the chemical forms of heavy metals in contaminated soil of this area [23,24]. They reported that the mentioned biochar and its combination with AP could significantly diminish the exchangeable fraction of Pb and Zn with an application rate of 5% and 10%. Overall, the effect of biochar produced at 400 °C and 600 °C on reducing the exchangeable fraction of Pb and Zn in contaminated soil had no significant or slight difference. Therefore, the present study is the first study to investigate and compare the effects of biochar produced from peanut shells and corn cob at 300 °C and their combination with apatite ore on chemical fractions of Pb and Zn in a multiple-metal contaminated field soil located near a Pb/Zn mine, in Vietnam. The present study hypothesized that the biochar produced at 300 °C and its combination with apatite might also decrease or immobilize the Pb and Zn's exchangeable fraction in the studied soil, the same as biochar made at 400 °C and 600 °C. If it could immobilize heavy metals as efficiently as the same biomass biochar produced at 400 °C and 600 °C, it would help the farmers save significant energy to produce large-scale biochar for incubation in practical field soil. If it could not diminish the exchangeable fraction of Pb and Zn as efficiently as the same biomass biochar produced at 400 °C and 600 °C, the result would provide valuable data for studying the effects of biochar made from different biomasses, such as peanut shells and corn cob, on the chemical fractions of heavy metals. This kind of this study is necessary since the number of studies on the impact of the combination of biochar with apatite on heavy metals' chemical fractions has been limited, especially on biochar produced from various agricultural biomasses.

This study aimed to investigate the impacts of biochar produced from agricultural wastes, including peanut shells and corn cob, at 300 °C, and their combination with apatite ore on the heavy metals' chemical fractions. The present study hypothesized that biochar made from peanut shells and corn cob and their blend with apatite could significantly diminish the exchangeable fraction of heavy metals in contaminated soil by converting them from mobile to immobile fractions. Thus, the present study was conducted to ascertain (i) the properties of corn-cob and peanut-shell-derived biochar pyrolyzed at 300 °C (PB300 and CB300); (ii) the alterations in soil properties after a 30-day incubation with biochar and apatite at the application

rates of 3% and 5%; (iii) the impacts of biochar and the blend of biochar–apatite on the chemical speciation of Pb and Zn, especially the exchangeable fraction, for remediating heavy metals in contaminated field soil; and (iv) if there were any significantly different effects between PB300 and CB300 on reducing the heavy metal's exchangeable fraction. This study was carried out in the laboratory to have initial results, which might be a significantly quick test before incubating a combination of peanut shell and corn cob biochar produced at 300 °C with apatite in an actual contaminated cornfield soil in a long-term experiment (6 months, 12 months, and 36 months) for immobilizing lead and zinc. The present study's results might contribute significantly to solving heavy metal pollution in soil and sustaining soil.

## 2. Materials and Methods

### 2.1. Soil Samples and Amendments

- Soil samples

The cornfield soil sample was surface soil collected at 0–20 cm depth in a lead and zinc mine in Thai Nguyen City, Vietnam (21°43′46.27″ N; 105°51′2.75″ E).

- Biochar

Peanut shells and corn cob were purchased in the markets in Thai Nguyen City and washed with deionized water. After that, they were dried in the oven at 45 °C for two days. Then they were pyrolyzed in a kiln stove [20] at 300 °C in an oxygen-limited condition for one hour. The biochar produced from peanut shells and corn cob at 300 °C were coded as PSB300 and CCB300. The produced biochar PSB300 and CCB300 were dried at 105 °C and then cooled and stored in plastic zip bags for further analysis and experiments.

- Apatite

Apatite ore was bought in Lao Cai province in Vietnam, and the supplier of apatite ore was Vietnam Apatite Limited company (22°29′8.02″ N; 103°58′14.38″ E). Before adding to the studied soil as the amendments, biochar (PSB300 and CCB300) and apatite were crushed to a size smaller than 1 mm and dried in the oven at 45 °C for three days. Then, transferred to plastic zip bags and stored in a cool and dry place for further analysis.

### 2.2. Design of Experiments

Different ratios of apatite and biochar were added to the soil sample. Apatite and biochar were combined with the studied soil at a mass ratio of 3% and 5% of biochar and 3:3% of biochar/apatite (*w*/*w*) and then mixed thoroughly to create a homogeneous mixture. Six incubations were carried out, each with three identical plots for each modification. In addition, three unaltered control incubations were performed to compare to amended soils. There was a total of 21 microcosm incubation plots. Each microcosm plot was analyzed in three replicates. Hence, the whole 63 runs were performed for the experimental procedure. The experiment's setup and description are provided in detail in Table 1. Distilled water was supplied during the incubation to maintain the soil's moisture at roughly 70% [20,23]. The soil samples were incubated for 30 days before being dried in the oven at 45 °C for three days and powdered to a size of less than 2 mm for later analysis [23,24].

**Table 1.** Designation of incubation experiment of amendments into the studied contaminated soil.

| Sample Plot | Sample Code | Ratio (%) |
|---|---|---|
| CS | CS | 0 |
| CS + 3% PSB300 | PSB3:3 | 3 |
| CS + 5% PSB300 | PSB3:5 | 5 |
| CS + 3% PSB300 + 3% AP | PSB3A3 | 3:3 |
| CS + 3% CCB300 | CCB3:3 | 3 |
| CS + 5% CCB300 | CCB3:5 | 5 |
| CS + 3% CCB300 + 3% AP | CCB3A3 | 3:3 |

PSB300: Peanut-shell-derived biochar produced at 300 °C; CCB300: Corn-cob-derived biochar made at 300 °C; AP: apatite; CS: control soil; incubation time: 30 days; 3 replicates, 21 experimental plots, 63 runs in total.

### 2.3. Analysis Methods of Soil and Materials

2.3.1. Physicochemical Properties Analysis

The fundamental physicochemical characteristics of the soil samples and amended materials (PSB300, CCB300, and AP) before and after the 30-day incubation with amendments were examined. Values of pH and EC of the soil, biochar, and apatite were determined using a Hanna HI 9124 pH meter (Rumani). The pH values of soil and materials were combined in a 1:10 ($w/v$) ratio with KCl 1M, stirred, and allowed to stand for one hour before measuring [24]. The pipette method was used to examine the texture of the studied soil [24,30]. The C/N multi 3100 (Analytik Jena, Jena, Germany) was utilized to analyze the material's organic carbon (OC) content in biochar and apatite [31].

2.3.2. Heavy Metal Analysis

According to EPA 3051A [32], soil samples were digested in a microwave with aqua regia to determine the heavy metals concentration. In a nutshell, 0.5000 g of material was weighed, 8 mL of concentrated $HNO_3$:HCl ($v/v$ = 1:3) was added, and the Mars 6 microwave system was used for digestion. The microwave system's operating parameters are shown in Table S1 (see Supplementary Materials). The digested solutions were cooled and then transferred to a Whatman No. 42 filter in order to filter the solution. After filtration, the filter solution was transferred into a 100 mL flask, diluted with deionized water, and stored in the fridge for further analysis [24].

The ICP-MS (Agilent 7900, Agilent Technologies, 5301 Stevens Creek Blvd, Santa Clara, CA, USA) was used to determine the concentration of heavy metals. Table S2 (see Supplementary Materials) illustrates the ICP-MS's operational parameters. The reference values of the heavy metal content of the sediment standard reference material (MESS-4) were compared to evaluate the recovery of heavy metals in the analysis process. The reference material (MESS-4) was repeatedly analyzed three times to determine the total concentration of heavy metals. The recovery results showed that the Pb and Zn's recovery rates for MESS-4 were 109.27% and 103.22%, respectively, as reported in the previous study [23] (see Table S2 in Supplementary Materials).

Tessier's sequential extraction method was used to determine the chemical fractionations of Pb and Zn [29], which classified the chemical speciation into five main fractions: exchangeable (F1), carbonate bound (F2), Mn/Fe-hydroxide bound (F3), organic substance bound (F4), and residue (F5). The detail of Tessier's sequential extraction procedure is shown in Table S3 (see Supplementary Materials) [28].

2.3.3. Surface Characteristics of Biochar and Apatite

Energy dispersive spectroscopy (EDS), field emission electron microscopy (FE-SEM), surface area and dimensional pore analysis (BET analyzer), and Fourier transform infrared spectroscopy (FTIR) were applied to evaluate the fundamental features of the surface morphology of materials. The functional groups of the surface's materials were examined using Fourier transform infrared spectroscopy (FTIR, JASCO FT/IR-4600, JASCO International Co. Ltd., Tokyo, Japan) [24]. Analysis of the surface morphology and chemical components of the biochar and apatite biochar was performed using a field emission electron microscope (FE-SEM, JSM-6700F, JEOL, Akishima Tokyo, Japan) equipped with an energy dispersive spectrometer (EDS) [24]. To investigate the dimensional pores and surface area of biochar and apatite, a BET analyzer (TriStar II 3020, Micromeritics Instrument Corporation, 4356 Communications Dr, Norcross, GA 30093, USA) was used [24].

### 2.4. Data Analysis and Statistics

Origin Pro 2021 (OriginLab Corp., Northampton, MA, USA) and Excel 2019 (Microsoft) software were used to analyze the data. The standard deviation and average values of the triple-result findings were analyzed using Excel 2019. The tables and graphs gave the data as mean value standard deviation. Using Origin Pro 2021, one-way ANOVA was performed to determine how the mean values varied between the treatments. A *p*-value of

0.05 or below was deemed significant. Spearman correlation was conducted to investigate the correlation between the factors using Origin Pro software 2021, version 9.8.

## 3. Results and Discussion

### 3.1. Physicochemical Properties of the Studied Soil and Materials

The previous study already examined and reported the soil texture [23]. The results of the basic properties of the studied soil are illustrated in Table 2.

**Table 2.** Physical and chemical characteristics of amendments and soil (mean ± standard deviation; three replicates n = 3).

| Properties | Unit | Studied Soil | CCB300 | PSB300 | AP |
|---|---|---|---|---|---|
| Sand | % | 69.78 ± 0.72 | - | - | - |
| Silt | % | 5.48 ± 0.32 | - | - | - |
| Clay | % | 24.74 ± 0.43 | - | - | - |
| pH | | 6.65 ± 0.01 | 9.53 ± 0.01 | 9.50 ± 0.01 | 8.99 ± 0.01 |
| OC | % | 1.95 ± 0.31 | 81.29 ± 0.12 | 75.82 ± 0.31 | 2.21 ± 0.10 |
| EC | $\mu S\ cm^{-1}$ | 119.20 ± 0.30 | 1675.40 ± 1.70 | 1115.30 ± 1.41 | 380.71 ± 0.90 |
| Pb | $mg\ kg^{-1}$ | 2973.77 ± 33.23 | <LOD | <LOD | <LOD |
| Zn | $mg\ kg^{-1}$ | 2462.24 ± 34.29 | 0.20 ± 0.02 | 0.21 ± 0.04 | 7.53 ± 0.02 |
| $S_{(BET)}$ | $m^2\ g^{-1}$ | - | 1.73 | 23.41 | 0.43 |

OC: organic carbon; EC: electrical conductivity; PSB300: Peanut-shell-derived biochar produced at 300 °C; CCB300: Corn-cob-derived biochar produced at 300 °C; AP: apatite ore; $S_{(BET)}$: surface area, LOD: limit of detection, -: no analysis.

The studied soil showed a low OC concentration with a mean value of 2.49 ± 0.12% and an EC's mean value of 136.5 ± 0.5 $\mu S\ cm^{-1}$, respectively. The soil sample had a pH of 6.65, which indicated that it was a mildly acidic soil type. The soil sample contained the average content of Pb and Zn of 2973.77 ± 33.23 $mg\ kg^{-1}$ and 2462.24 ± 34.29 $mg\ kg^{-1}$, respectively. These figures were approximately 42 and 12 times higher than the acceptable standard (70 and 200 $mg\ kg^{-1}$ for Pb and Zn, respectively) set by the Vietnamese Regulation (2015) [33], demonstrating that the studied soil was exclusively polluted with lead and zinc. The contents of Pb and Zn in the studied soil were superior and dominant compared to other heavy metals such as Cu, Cd, Cr, As, and Ni (data not shown). Hence, this study focused only on the chemical fractions of Pb and Zn in the investigated soil.

The fundamental physicochemical properties of CCB300, PSB300, and apatite are also illustrated in Table 2. The results in Table 2 show that the pH values of CCB300, PSB300, and apatite were 9.53 ± 0.01, 9.50 ± 0.01, and 8.99 ± 0.01, respectively. These pH values were substantially higher than the studied soil, suggesting that the soil's pH could be raised after being incubated with biochar and apatite. Because of the breakdown of acidic groups (-COOH and -OH), carbonate formation, and the alkali salts' separation from organic molecules as the pyrolysis temperature increased, biochar's high pH values were attributed to these processes [34,35]. Furthermore, CCB300, PSB300, and AP had higher EC values than the studied soil, which might increase the EC value of the contaminated soil after the incubation, facilitating the ion exchange reactions that occurred in the soil solution [24].

### 3.2. Characteristics of Amendments

3.2.1. Fourier Transform Infrared Spectroscopy Analysis of Amendments (FT-IR)

- FT-IR of biochar

The IR spectra of pristine corn cob (CC), peanut shells (PS), and biochar (CCB300 and PSB300) are illustrated in Figure 1A,B. Overall, the CC and PS's infrared spectra were similar with the primarily noticeable peaks at about 3428, 2921, 2364, 1640, 1038, and 608 $cm^{-1}$. While the CCB300 and PSB300 had identical peaks at around 3428, 2921, 2364, and 1640 $cm^{-1}$. These peaks were indistinguishable from those in the pristine materials' spectra. However, the peak intensity in the biochar's spectra was considerably lower than that of pristine materials (CC and PS). The prominent peak at ~3428 $cm^{-1}$ in the spectra of CC, PS, CCB300, and PSB 300 was ascribed to the hydroxy (O-H) [36,37]. The intensity of this peak

in the biochar's spectra was much weaker than that of pristine materials, indicating that the hydroxy group had been significantly decomposed during the pyrolysis process. The C-H stretch vibration was attributed to the peak occurring at about 2921 cm$^{-1}$ [38,39]. The peak at around 2364 cm$^{-1}$ can be associated with the C≡C stretch alkynes functional group [40]. The pronounced peak at about 1638–1640 cm$^{-1}$ was referred to as the C=O stretching and aromatic C=C vibrations [39,41], while the peak at around 1038 cm$^{-1}$ can be the presence of C-O-H or C-O-C stretching or allopathic [36,42]. Lastly, the peak at about 608 cm$^{-1}$, being noticeable in CC and PS, but negligible in CCB300 and PSB300, can be assigned to a C-OH out-of-plane bending mode of aromatic compounds [43]. In conclusion, the IR spectra of CCB300 and PSB300 indicated the hydroxyl group in cellulose, the carbonyl groups in the acetyl ester in hemicellulose, and the carbonyl aldehyde in lignin were the primary causes of all absorption bands [23,40]. These functional groups can play a crucial role in immobilizing heavy metals in contaminated soil through the exchange or complex reactions [24].

- FT-IR of apatite

The IR spectra of AP are demonstrated in Figure 1C. The apatite ore's IR spectra had four noticeable peaks at around 1904, 1049, 574, and 464 cm$^{-1}$, which were assigned to characteristic bands of $PO_4^{3-}$, namely, the asymmetric stretching vibration of the P-O bond and the asymmetric bending vibration of the O-P-O bond [23]. The OH group's vibration or the water absorbed was associated with the two pronounced peaks at ~3618 and ~3446 cm$^{-1}$ [23,44,45]. The presence of the F-ion was ascribed to the peak at about 797 cm$^{-1}$ [23]. Furthermore, there was a noticeable peak at around 1437 cm$^{-1}$, associated with $CO_3^{2-}$ stretching vibration [46,47]. The FT-IR analysis revealed that the primary functional groups in apatite ore were $PO_4^{3-}$, -OH, or $CO_3^{2-}$, which can precipitate or exchange with heavy metals to form new compounds in the soil solution during the incubation [23]. The FT-IR result of the present study was consistent with the conclusion informed by previous studies [23,45–47], which concluded that the studied apatite ore was a fluor-hydroxide-carbonate-apatite.

### 3.2.2. SEM-EDS Analysis of Amendments

- SEM analysis of materials

SEM results of pristine corn cob, peanut shell, apatite, and biochar (CCB300 and PSB300) are illustrated in Figure 2. The SEM images demonstrate that pristine corn cob (Figure 2A), peanut shell (Figure 2B), and apatite (Figure 2C) were not wholly homogeneous but had a somewhat flat surface without holes. Whereas CCB300 (Figure 2D) and PSB300 (Figure 2E) had porous and hollow surfaces, which might facilitate the adsorption of heavy metals on the surface of biochar.

- EDS analysis of biochar and apatite ore

In addition to the SEM analysis, the chemical elements of the materials' surfaces were investigated using EDS equipped with SEM. The EDS results of biochar and apatite ore are shown in Figure 3. The primary chemical elements of CCB300 were C and O, with about 78.64% and 12.09% of the weight. Moreover, CCB300 had various inorganic elements such as K, P, Si, and Mg, which might benefit heavy metals' exchange or precipitation reactions when applied in contaminated soil. Likewise, PSB300 had C and O as the main chemical elements, with 79.81% and 11.25%, respectively. Additionally, it had diverse inorganic elements such as N, Si, Al, P, Cl, Mg, Zn, Fe, Ca, and K, which might enable the exchange and precipitation reactions of heavy metals in the polluted soil.

In the meantime, apatite ore had the main component of O, Si, Mg, and C with, respectively, 52.99%, 12,36%, 6.24%, and 6.24% of the weight. Moreover, apatite had other elements, such as Al, K, Ca, N, and P, which might involve the immobilization process of heavy metals in the soil solution.

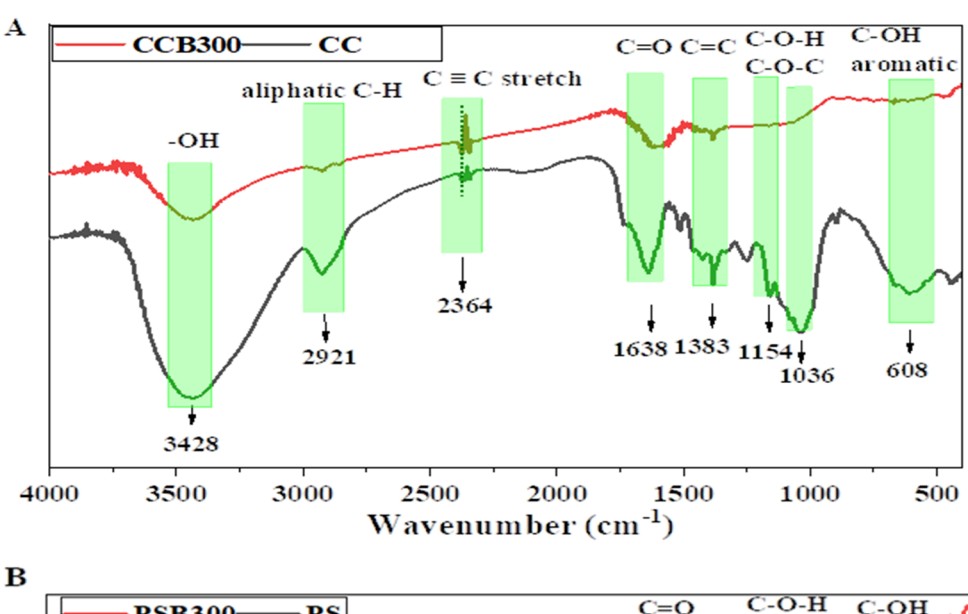

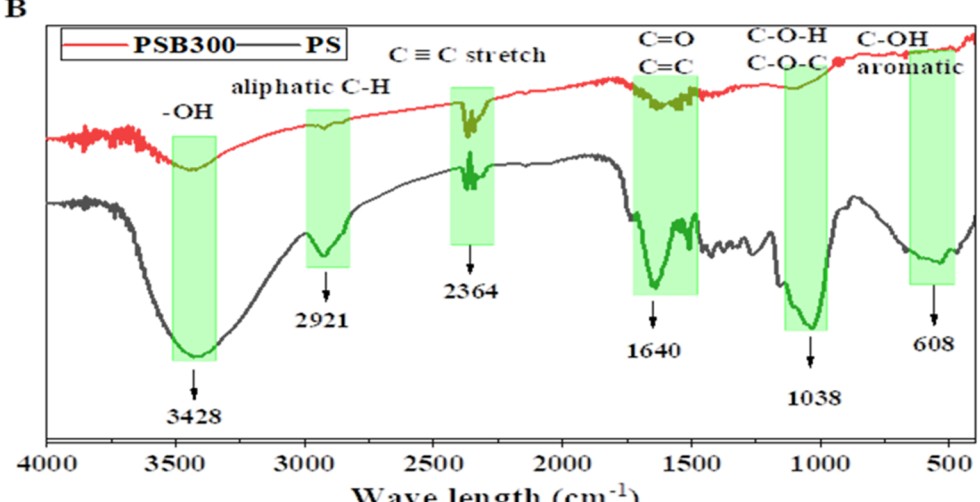

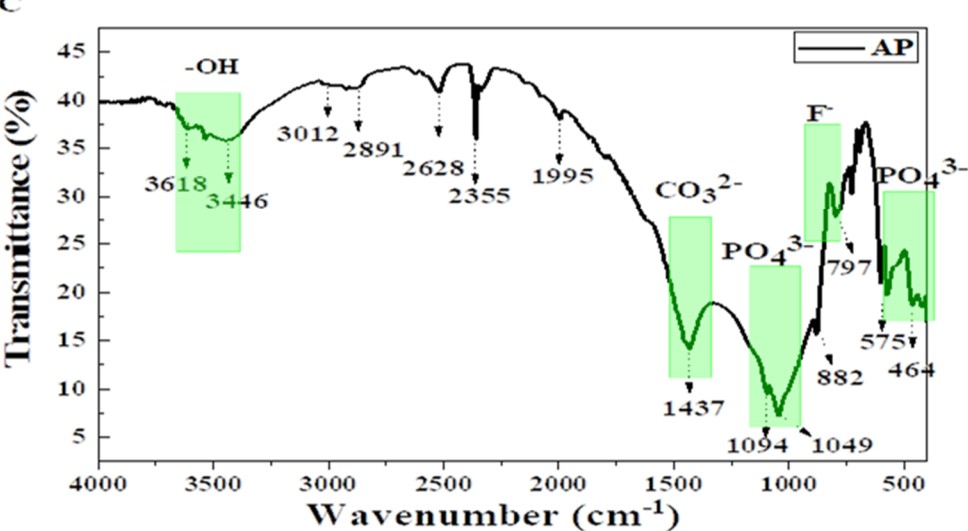

**Figure 1.** (**A**) FT-IR spectra of corncob (CB) and its biochar produced at 300 °C (CCB300); (**B**) FT-IR spectra of peanut shell (PS) and its biochar produced at 300 °C (PSB300); (**C**) FT-IR spectra of apatite ore (AP).

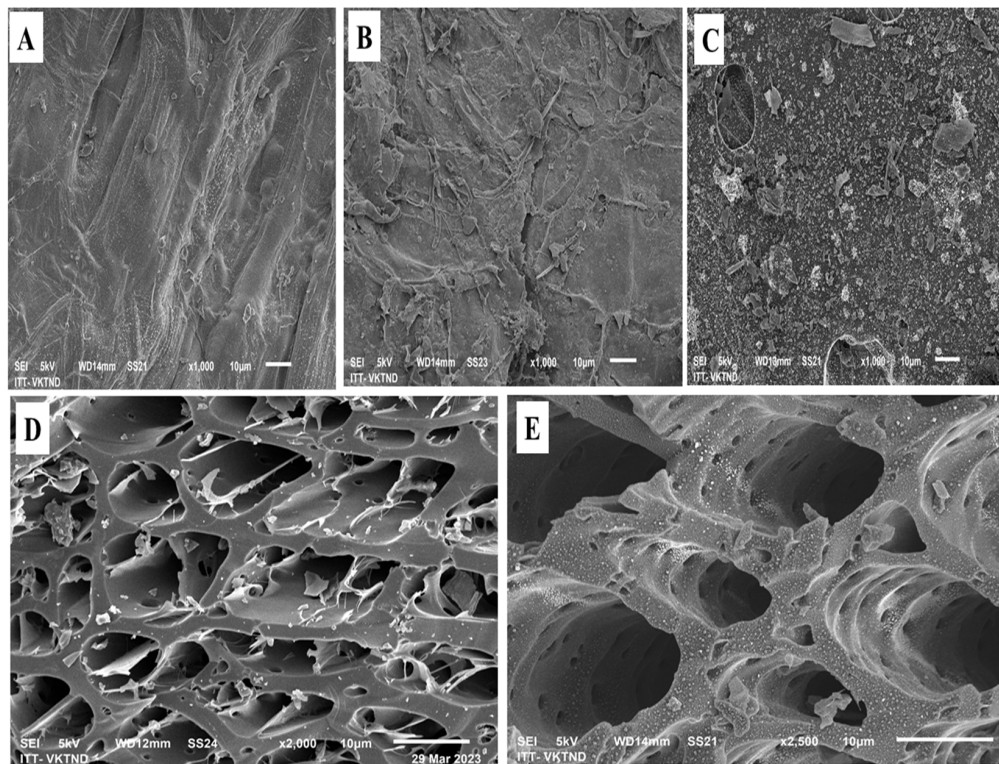

**Figure 2.** SEM images of (**A**) Corn cob, (**B**) Peanut shell, (**C**) apatite ore (AP), (**D**) peanut-shell-derived biochar produced at 300 °C (PSB300), (**E**) corn-cob-derived biochar produced at 300 °C (CCB300).

In sum, the SEM results showed that biochar CCB300 and PSB300 had porous and hollow surfaces, while AP had a slightly flat surface, indicating that biochar might be superior in the physical adsorption of heavy metals on their surfaces than AP. Furthermore, biochar and AP had various elements that might play a vital role in immobilizing heavy metals in contaminated soil.

*3.3. Alteration of OC, pH, and EC of the Incubated Soil after a 30-Day Incubation*

The summary of pH, organic carbon content (OC), and electrical conductivity (EC) in the incubated soil samples after one-month incubation with amendments is shown in Table 3. The changes in pH, OC, and EC of the incubated soils are described in the following sections.

**Table 3.** Exchangeable fraction of Pb, Zn, OC, pH, and EC following 30-day biochar (PSB300, CCB300) and apatite incubation (mean ± standard deviation; three replicates n = 3).

| Sample | F1_Pb | F1_Zn | pH | OC | EC |
|---|---|---|---|---|---|
| | mg kg$^{-1}$ | | | g kg$^{-1}$ | µS/cm |
| CS | 578.9 ± 10.5 a | 375.0 ± 12.6 a | 6.65 ± 0.01 c | 20.1 ± 0.3 e | 119.1 ± 1.3 g |
| PSB3:3 | 415.6 ± 9.5 e | 259.9 ± 11.8 c | 6.74 ± 0.01 b | 42.1 ± 0.4 d | 150.4 ± 0.3 f |
| PSB3:5 | 429.7 ± 6.8 e | 271.8 ± 13.2 c | 6.79 ± 0.01 a | 56.9 ± 0.1 b | 171.2 ± 0.3 c |
| PSB3A3 | 491.1 ± 8.7 c | 278.7 ± 10.5 c | 6.81 ± 0.01 a | 41.5 ± 0.5 d | 158.2 ± 0.2 e |
| CCB3:3 | 502.7 ± 10.4 bc | 315.3 ± 14.2 b | 6.74 ± 0.01 b | 44.4 ± 0.9 c | 165.7 ± 0.7 d |
| CCB3:5 | 468.2 ± 8.9 d | 277.2 ± 15.2 c | 6.79 ± 0.01 a | 60.7 ± 0.4 a | 196.8 ± 0.8 a |
| CCB3A3 | 519.5 ± 9.3 b | 291.1 ± 9.7 c | 6.81 ± 0.01 a | 43.8 ± 0.9 bc | 173.5 ± 0.4 b |

According to the *t*-test, the same letter indicates a non-significant difference ($p < 0.05$) between the two fields; the alphabet letters (a, b, c, d, e, f, g) were arranged from high to low value of the mean values to point out the different values.

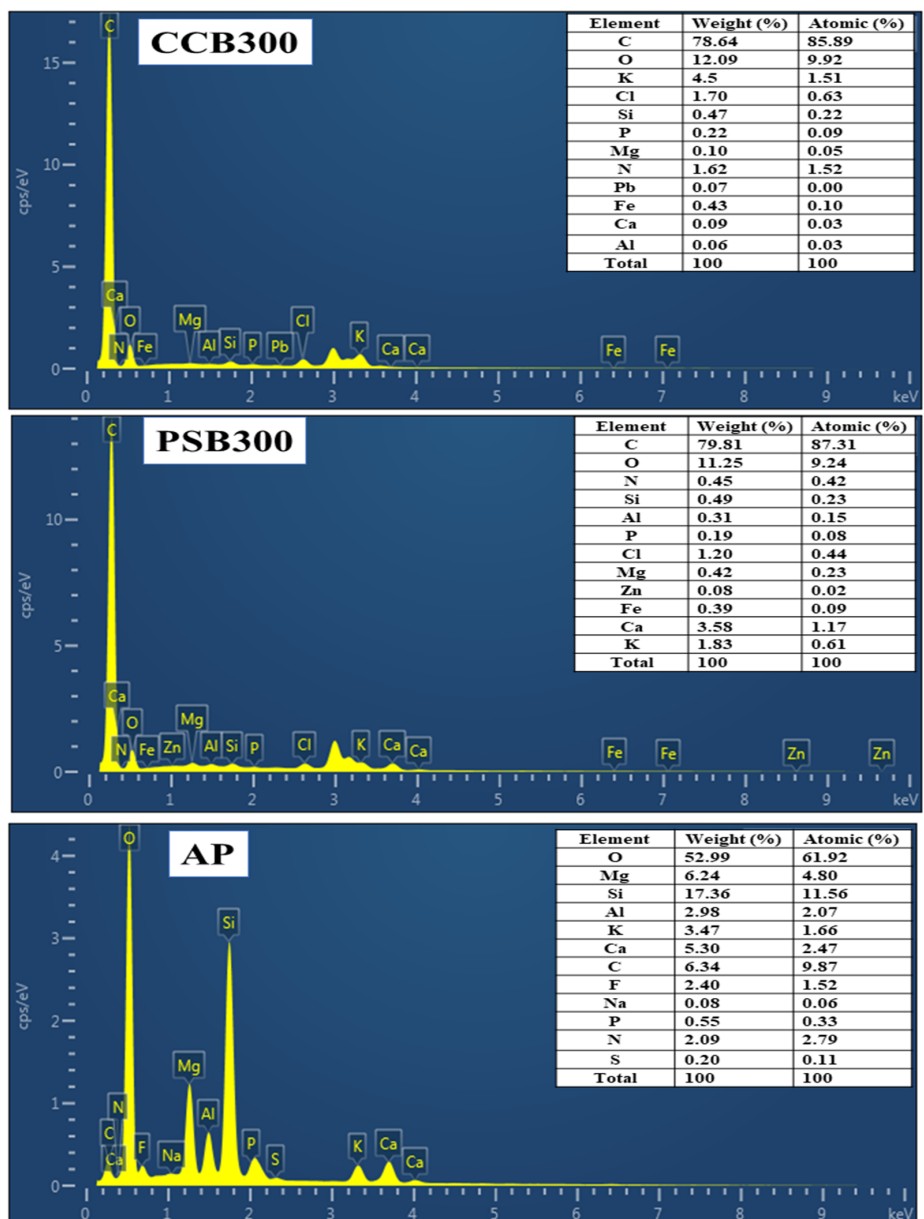

**Figure 3.** EDS images of corn-cob-derived biochar produced at 300 °C (CCB300), peanut-shell-derived biochar created at 300 °C (PSB300), and apatite ore (AP).

### 3.3.1. Soil pH

The pH value of the control soil (CS) was 6.65 ± 0.01. After the 30-day incubation, the pH values of incubated soil increased significantly compared to that of CS ($p < 0.05$). The pH values of incubated soils ranged from 6.74 to 6.81 and increased with amended application rates. The pH value was highest in samples PSB3:5, PSB3A3, CCB3:5, and CCB3A3 when the incubation ratios were 5% biochar and 3:3% of biochar/apatite. The increased pH values in the incubated soil were attributed to the high pH value of biochar and AP compared to the studied soil. The increase in soil pH after incubation with amendments, such as biochar was also reported in previous studies [20,23,24].

### 3.3.2. Organic Carbon (OC)

The control soil had an organic carbon content of 20.1 ± 0.3 g kg$^{-1}$, while this figure in the incubated soil after one month of incubation rose significantly compared to the control ($p < 0.05$) soil. The most significant increase in the OC value was in the sample CCB3:5

with $60.7 \pm 0.4$ g kg$^{-1}$ when the biochar application rate was 5%. The OC value of the soil sample PSB3:5 ranked second with $56.9 \pm 0.1$ g kg$^{-1}$. The rising in the OC value of incubated soils was associated with the high OC value of PSB300 (75.82%) and CCB300 (81.29%) (see Table 1). The CCB300 had a higher OC value than the PSB300, explaining why the CCB3:5 had the most considerable OC value while the PSB3:5 ranked second. Previous studies also reported a significant increase in OC values of incubated soil after the incubation with biochar [15,23,24,48].

### 3.3.3. Electrical Conductivity (EC)

The EC value in the control soil was $119.1 \pm 1.3$ µS cm$^{-1}$, while this value of incubated soil increased considerably compared to that of CS ($p < 0.05$, significant difference). The EC values increased with the amendment application rates, and the high EC values of CCB300 and PSB300 could explain this result. The EC values in soil samples incubated with CCB300 were slightly higher than those of soil samples incubated with PSB300, and the EC value was highest in the soil sample CCB3:5 ($196.8 \pm 0.8$ µS cm$^{-1}$) when the soil was incubated with CCB300 at a ratio of 5%. The higher EC value of CCB300 than PSB300 led to this phenomenon. The finding of this study was in agreement with the results reported by previous studies [23,24], which informed that the EC of the incubated soil increased with the biochar application rates.

### 3.4. Chemical Speciation of Lead and Zinc in Control and Incubated Soils after a 30-Day Incubation

#### 3.4.1. Pb Speciation

The chemical fractions of lead (F1-Pb) in the blank soil (CS) and incubated soils are shown in Table 4. And the proportion of Pb and Zn's chemical fractionations in soil are illustrated in Table S4 and Figure 4. Overall, the chemical fractionations of lead distributed in the decreasing order of F2 > F5 > F1 > F3 > F4, and the F2-Pb contributed almost 50% of the total, while the exchangeable fraction of Pb (F1-Pb) contributed approximately nearly 20% of all fractions (Figure 4A), about 578.9 mg kg$^{-1}$, which might pose a severe threat to the surrounding environment.

**Table 4.** The concentration of Pb and Zn in fractions in soil samples after 30 days incubated with biochar (PSB300 and CCB300) and apatite (mean $\pm$ standard deviation; three replicates n = 3).

| Metal | Sample | F1 | F2 | F3 | F4 | F5 |
|---|---|---|---|---|---|---|
| | | (mg kg$^{-1}$) | | | | |
| | CS | $578.9 \pm 15.5$ a | $1376.1 \pm 23.1$ b | $344.1 \pm 9.5$ c | $129.4 \pm 3.7$ c | $549.2 \pm 10.1$ e |
| | PSB3:3 | $415.6 \pm 9.5$ e | $1384.7 \pm 29.3$ b | $372.3 \pm 6.7$ a | $138.9 \pm 5.8$ b | $592.1 \pm 6.4$ c |
| | PSB3:5 | $429.7 \pm 6.8$ e | $1394.7 \pm 35.4$ b | $338.0 \pm 7.5$ c | $151.2 \pm 2.3$ a | $646.4 \pm 5.3$ a |
| Pb | PSB3A3 | $491.1 \pm 8.7$ c | $1345.6 \pm 26.7$ b | $365.4 \pm 8.5$ b | $130.9 \pm 4.4$ c | $613.4 \pm 6.2$ b |
| | CCB3:3 | $502.7 \pm 11.4$ bc | $1387.8 \pm 47.1$ b | $372.3 \pm 9.3$ a | $139.7 \pm 3.1$ b | $572.1 \pm 9.5$ d |
| | CCB3:5 | $468.2 \pm 8.9$ d | $1458.2 \pm 19.3$ a | $322.8 \pm 5.6$ d | $154.2 \pm 4.2$ a | $602.0 \pm 10.7$ bc |
| | CCB3A3 | $519.5 \pm 9.3$ b | $1467.4 \pm 32.8$ a | $319.7 \pm 9.1$ d | $136.6 \pm 5.3$ bc | $556.4 \pm 12.8$ de |
| | CS | $375.0 \pm 12.6$ a | $647.0 \pm 19.7$ b | $788.2 \pm 12.4$ a | $33.0 \pm 2.3$ b | $669.8 \pm 8.7$ d |
| | PSB3:3 | $259.9 \pm 11.8$ c | $698.7 \pm 18.4$ a | $676.0 \pm 8.3$ c | $37.3 \pm 3.1$ b | $705.1 \pm 15.8$ c |
| | PSB3:5 | $271.8 \pm 13.2$ c | $716.0 \pm 12.8$ a | $784.7 \pm 9.0$ a | $47.2 \pm 3.2$ a | $737.3 \pm 16.3$ b |
| Zn | PSB3A3 | $278.7 \pm 10.5$ c | $715.8 \pm 17.2$ a | $730.8 \pm 11.2$ b | $35.2 \pm 2.7$ b | $715.2 \pm 14.2$ bc |
| | CCB3:3 | $315.3 \pm 14.2$ b | $645.7 \pm 18.3$ b | $684.7 \pm 4.5$ c | $36.9 \pm 3.4$ b | $735.1 \pm 11.2$ b |
| | CCB3:5 | $277.2 \pm 15.2$ c | $703.5 \pm 19.3$ a | $645.5 \pm 6.1$ d | $44.5 \pm 1.3$ a | $794.8 \pm 18.3$ a |
| | CCB3A3 | $291.1 \pm 9.7$ c | $710.8 \pm 12.8$ a | $639.5 \pm 4.3$ d | $44.1 \pm 4.5$ a | $737.3 \pm 21.2$ b |

According to the $t$-test, the same letter indicates a non-significant difference ($p < 0.05$) between the two fields. The alphabet letters (a, b, c, d, e) were arranged from high to low value of the mean values to point out the different values.

- Exchangeable fraction (F1-Pb)

The control soil (CS) had an F1-Pb value of $578.9 \pm 15.5$ mg kg$^{-1}$ and this figure for incubated soils was significantly less than that of CS ($p < 0.05$, significant difference). The decrease in F1-Pb in the treated soil varied with the amendments. In the soil incubated with biochar PSB300, the value of F1-Pb in sample PSB3A3 (when the ratio of biochar: apatite = 3:3%) was $491.1 \pm 8.7$, decreasing about 15% compared to that of CS, when

these figures in samples PSB3:3 and PSB3:5 were not significantly different ($p > 0.05$, non-significant difference), and they declined up to approximate 28% compared to that of CS, illustrating that F1-Pb did not change significantly when the PSB300 incubation ratios increased from 3% to 5% after 30-day incubation, but the combination of PSB300 and apatite with a proportion of 3:3% had less impact in diminishing F1-Pb compared to only biochar application at the same rate of 3%. Meanwhile, the F1-Pb values in soil samples incubated with CCB300 also significantly declined compared to CS, but the decreasing level was less than in soil samples incubated with PSB300 ($p < 0.05$, significant difference). The most significant reduction in F1-Pb in the soil incubated with CCB300 was in sample CCB3:5, with about 19%. In addition, the F1-Pb values in the samples CCB3:3 and CCB3A3 were the same, while that in the sample CCB3:5 was less than in CCB3:3 and CCB3A3, indicating that the exchangeable fraction of Pb decreased when the CCB300 application ratio rose. The results showed slightly diverse impacts on reducing the Pb exchangeable fraction in the incubated soils. The significant decrease in F1-Pb after the incubation of biochar and biochar/apatite was reported in previous studies [20,23,24].

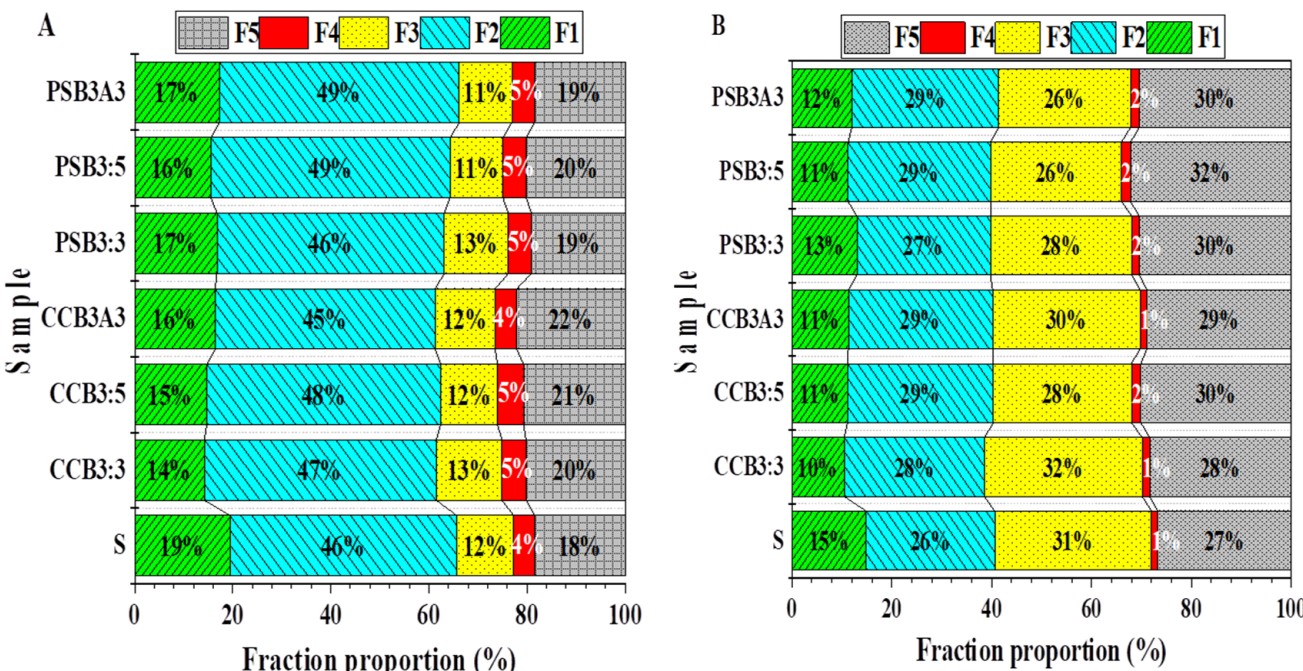

**Figure 4.** Proportions of chemical fractions of Pb (**A**) and Zn (**B**) in soil samples after the 30-day incubation.

- Carbonate fraction (F2-Pb)

Pb's carbonate fractionation (F2-Pb) in the control soil was $1376.1 \pm 23.1$ mg kg$^{-1}$. This figure was not significantly different in samples PSB3:3, PSB3:5, and PSB3A3 ($p > 0.05$, non-significant difference) when soil incubated with biochar PSB300 at the ratio of 3% and 5% and the mixture of PSB300/AP (3:3%). In contrast, after being incubated with CCB300, sample CCB3:3 had no significant change in F2-Pb in comparison to CS ($p > 0.05$, non-significant difference), but there were considerable changes in the F2-Pb in samples CCB3:5 and CCB3A3 ($p < 0.05$, significant difference) when the F2-Pb in samples CCB3:5 and CCB3A5 increased significantly compared to CS ($p < 0.05$, significant difference). A previous study reported that after incubation with 3%, 5%, or 3:3% of rice straw biochar/apatite, the F2-Pb in the incubated soil increased considerably compared to that of CS [20].

- Fe/Mn-Oxide fraction (F3-Pb)

The F3-Pb value of the control soil was $344.1 \pm 9.5$ mg kg$^{-1}$. This figure increased slightly in samples PSB3:3 and CCB3:3 ($p < 0.05$), but did not change significantly in samples

PSB300; however, this figure in samples CCB3:5 and CCB3A3 decreased slightly when the soil was incubated with CCB300 at a ratio of 5% and a mixture of CCB300/AP at a ratio of 3:3%. Previous studies reported mixed results of the change in F3-Pb when the soil was incubated with biochar. Dang et al. (2019) informed that after being treated with biochar produced from rice straw at the ratio of 3% and 5%, there was no significant change in F3-Pb compared to the control soil, while that figure decreased significantly in the soil sample incubated with the mixture of biochar/apatite at the ratio of 3:3% [20]. Additionally, Awad et al. (2021) reported that there was no significant change when incubated with garden waste-derived biochar at the 2, 4, and 6% ratios [48], while the biochar produced from the paulownia and bamboo amended with an application rate of 2%, 4%, and 6% could reduce this fraction slightly but significantly ($p < 0.05$, significant difference) [15].

- Organic carbon fraction (F4-Pb)

The figure for F4-Pb in the control soil was $129.4 \pm 3.7$ mg kg$^{-1}$, while it increased significantly in incubated soil ($p < 0.05$, significant difference), except in samples PSB3A3 and CCB3A3 when the soil samples were amended with a mixture of 3:3% of biochar/apatite. The F4-Pb value increased slightly with the increasing biochar application rate and was highest in the samples PSB3:5 and CCB3:5 when biochar PSCB300 and CCB300 were applied at the 5% ratio. The present result was consistent with earlier studies [14,23,48], which stated that the F4-Pb dramatically increased after biochar was incubated in contaminated soil compared to the control soil.

- Residual fraction (F5-Pb)

The control soil's lead residual fraction (F5-Pb) was $549.2 \pm 10.1$ mg kg$^{-1}$. This figure of incubated soils increased significantly compared to that of the control soil ($p < 0.05$, significant difference) and ranged from $556.4 \pm 12.8$ to $646.4 \pm 5.3$ mg kg$^{-1}$. The values of F5-Pb rose with the increasing biochar application rates. The soil samples incubated with PSB300 had slightly but significantly higher F5-Pb values than those of CCB300 at the same amended ratio. This result did not agree with earlier studies, which stated that the biochar incubation had no significant impact on the residual fraction of Pb. The mixed results may be attributed to the difference in biochar types, pyrolysis temperature, application rates, and soil types. For instance, studies conducted by Awad et al. [15] investigated biochar produced from garden wastes, paulownia biochar, and bamboo biochar at 700–800 °C, at the amended rates of 2, 4, and 6%, while Dang et al. (2019) investigated rice-straw-derived biochar, produced at 340–550 °C, with an application rate of 1, 3, and 5% to remediate heavy metals in the tailing soil [20]. This study investigated the utilization of biochar produced from peanut shells and corn cob at 300 °C at 3 and 5% amendment ratios to remediate Pb and Zn in agricultural soil.

### 3.4.2. Zn Speciation

Figure 4B and Table 4 illustrate the chemical speciation of zinc in control and incubated soils after 30-day incubation. The results revealed diverse alterations in zinc's chemical fractions of the amended soils. Overall, the chemical forms of zinc in soil distributed in the declining order of F5 > F2~F3 > F1 > F4 and the F5, F2, and F3 fractions of Zn contributed about one-third of the total, while the F1 was about one-tenth of the whole.

- Exchangeable fraction of zinc (F1-Zn)

The amount of zinc in the exchangeable fractionation of the control soil (CS) was $375.0 \pm 12.6$ mg kg$^{-1}$, and this value in incubated soil samples declined significantly when the amendment application rates rose compared to CS. This figure in the soil samples incubated with PSB300 had no significant change when amended with 3%, 5% of PSB300, or 3:3% of PSB300/apatite ($p > 0.05$). However, the quantity of F1-Zn in the soil treated with CCB300 changed slightly but significantly between soil samples incubated with 3% biochar (CCB3:3) and CCB3:5 and CCB3A3, while the soil samples amended with 5% CCB300 and a mixture of 3:3% of CCB300/AP had no significant difference ($p > 0.05$, non-significant

difference). PSCB300 and CCB300 had nearly identical effects on the decrease in F1-Zn, except for the sample CCB3:3.

Overall, the F1-Zn value in all incubated soils was diminished significantly in all amended soils, up to about 28% compared to CS, indicating that PSB300, CCB300, and AP could immobilize mobile zinc in contaminated soil after one-month incubation. The considerable reduction in F1-Zn in contaminated soil after the incubation with various biochar types, rates, and apatite was also reported in previous studies [19,48,49]

- Carbonate fraction of zinc (F2-Zn)

The control soil had a Zn carbonate fraction value of $647.0 \pm 19.7$ mg kg$^{-1}$ and this figure for incubated soils increased significantly in comparison with that of CS ($p < 0.05$, significant difference), aside from only sample CCB3:3 ($645.7 \pm 18$ mg kg$^{-1}$), which had no significant difference compared to CS ($p > 0.05$, no significant difference). Moreover, there were no significantly different effects on the F2-Zn fraction in the incubated soils between the two biochar types (PSB300 and CCB300), and between the rate of 5% and the mixture of 3:3% biochar/apatite ($p < 0.05$, significant difference). The increase in F2-Zn in the incubated soil after adding amendments was reported in a previous study [20], which studied rice straw biochar and apatite as amendments to immobilize heavy metals in tailing soil.

- Oxide bound fraction of zinc (F3-Zn)

The Fe/Mn oxide bound fraction of Zn (F3-Zn) in the control soil was $788.2 \pm 12.4$ mg kg$^{-1}$, and this figure for incubated soil samples decreased significantly compared to that of CS ($p < 0.05$, significant difference), except for the sample PSB3:5 ($p > 0.05$, no significant difference). There was no similar trend between the change in the F3-Zn of biochar PSB300 and CCB300. For the former, the more the application rate increased, the less F1-Zn was reduced, while for the latter, the more the application rate increased, the more F1-Zn was decreased. The effect of biochar 3% incubation and biochar/apatite 3:3% incubation of both PSB300 and CCB300 on the F3-Zn was significantly different for both biochar PSB300 and CCB300 ($p < 0.05$, significant difference). The significant decline in this fraction was also reported in previous studies [20,21].

- Organic compound fraction of zinc (F4-Zn)

The organic fraction of zinc (F4-Zn) in the control soil was $33.0 \pm 2.3$ mg kg$^{-1}$. After the incubation of biochar and apatite, this figure in the incubated soil increased significantly in the samples PSB3:5, CCB3:5, and CCB3A3 compared to that of the CS ($p < 0.05$, significant difference), while that figure in the sample PSB3:3, PSB3A3, CCB3:3 had no significant difference as compared to CS ($p > 0.05$, no significant difference). The rise of F4-Zn was associated with the high OC values of biochar PSB300 and CCB300. Overall, the higher the biochar application rate in the soil was, the higher the F4-Zn value was. This trend was similar to the results reported by previous studies [23,24].

- Residual fraction of zinc (Zn-F5)

The zinc concentration of the residual fraction in the control soil was $669.8 \pm 8.7$ mg kg$^{-1}$. After 30-day incubation with biochar and apatite, this figure of the incubated soils rose significantly compared to CS. CCB300 had more effects on the increase in F5-Zn than PSB300. Furthermore, the F5-Zn value increased with the biochar application rate. The addition of apatite at 3:3% with biochar showed no significant alteration in the F5-Zn of the incubated soil in comparison to the addition of only 3% biochar for bott PSB300 and CCB300 ($p < 0.05$, significant difference). The change in the F5-Zn in this study was quite different from the results reported in previous studies. Awad et al. (2021) informed that there was no significant change in this fraction after incubating with 2, 4, and 6% of biochar produced from paulownia and bamboo [15], while Dang et al. (2019) noticed that there was a significant decrease in F5-Zn in the soil after incubation with rice-straw-derived biochar at 3% and 5% and a mixture of biochar/apatite at a 3:3% ratio [20].

In general, the chemical fractions of lead and zinc in the incubated soil changed significantly in comparison with the control soil after being incubated with biochar (PSB300

and CCB300) and apatite for 30 days. The exchangeable fraction of Pb and Zn (F1-Pb and F1-Zn) in incubated soils altered the most in all fractions when compared to that of the control soil, diminishing F1-Pb and F1- Zn up to 26% and 33% for Pb and Zn, respectively, when calculated in proportion (Figure 4), approving the immobilization of Pb and Zn in the soil by biochar and apatite, at the application rates of 3% and 5%. In addition, there was a significant change in the F4 and F5 fractions, which are stable and rarely negatively impact the environment in natural conditions. This means that the incubation of biochar and apatite could partly convert the exchangeable fraction of heavy metals into stable fractions (F4 and F5) in the incubated soil. The three-percent biochar application rate could decrease the exchangeable fraction of Pb and Zn somewhat more than that of the five-percent application rate. However, the 5% amended ratio could slightly increase the proportion of Pb and Zn in the residual fraction more than that of the 3% amended ratio (Figure 4). The impacts of PSB300 and CCB300 on reducing the F1-Pb and F1-Zn were compared to the results of previous studies in Table 5.

At first glance, it seemed that PSB300 and CCB300 had less efficiency in reducing the exchangeable fraction of Pb and Zn in the studied soil when PSB300 and CCB300 could reduce the Pb and Zn's exchangeable fraction up to 26% and 33%, respectively, while the figures of other studies were 40.47% in reducing Pb in the labile fraction of biochar produced from sugarcane bagasse at 450 °C, or 57% and 47.5% in decreasing Pb and Zn in the exchangeable fraction by rice straw biochar produced at 450 °C. The different results might be attributed to the difference in soil types, pyrolyzed temperatures, and incubation time. For example, the soil type used in the study of Nie et al. (2018) [50] was 75% sand and 18% clay, while the field soil in this study had 69% sand and 24% clay. This means that the soil sample in the Nie et al. (2018) study had more sand and less clay than the present study's soil. Therefore, heavy metals might combine more tightly in the present study's soil than in the study of Nie et al. (2018), leading to diverse effects on the chemical forms of heavy metals. In addition, the incubation time in the present study was only one month, while it was two months in the study of Nie et al. (2018) and three months in the study of Dang et al. (2019) [20]. The more prolonged the incubation, the more biochar could reduce the exchangeable fraction of heavy metals. Dang et al. (2019) [20] reported that rice-husk-derived biochar could diminish the exchangeable fraction of Pb and Zn after a 90-day incubation better than a 30-day incubation. Therefore, it is hard to compare the effectiveness of PSB300 and CCB300 on reducing the exchangeable fraction of Pb and Zn with other biochars reported by previous studies when they were not conducted in the same experimental conditions.

However, CCB300 and PSB300 showed better effects when compared with the result of agricultural-waste-derived biochar in the study of Awad et al. (2021) [49] and biochar produced from bamboo and paulownia [15]. Although, in these previous studies, the incubation time was two months and longer than that of the present study (1 month) and the application rate was 6% and, therefore, higher than that of the present study (3% and 5%), the biochar in this study still could diminish the exchangeable fraction of Pb and Zn more than those in the previous studies reported by Awad et al. (2021).

Overall, it is hard to compare the effectiveness of various biochars in different studies since the reported experiments were not carried out under the same conditions. The comparison results in Table 5 indicate that biochar PSB300 and CCB300 could be potential materials for adsorbing and immobilizing Pb and Zn in contaminated soil by significantly reducing the exchangeable fraction of heavy metals (F1). Extended-time field experiments should be conducted to investigate the long-term effect of PSB300 and CCB300 and their combination with apatite on adsorbing Pb and Zn in contaminated soil.

**Table 5.** Comparison of the efficiency of various biochars in adsorbing and immobilizing the heavy metals' exchangeable fraction in contaminated soil.

| Biomass Feedstock | pH | Amended Rate | Incubation Time | Heavy Metals | Impacts | Ref |
|---|---|---|---|---|---|---|
| This study | PSB: 9.53, CCB: 9.5 | 3% and 5% | 1 month | Pb, Zn | Reducing 26% and 33% of the exchangeable fraction of Pb and Zn, respectively. | |
| sugarcane bagasse-derived biochar (produced at 450 °C) | SBC: 11.3 | 1.5, 2.25, and 3% | 2 months | Cu, Pb, Cd | Reducing 40.47% labile fraction of Pb. | [51] |
| garden waste biochar produced at 400 °C and 600 °C | BC400: 11 BC600: 11.5 | 2%, 4%, and 6% | 2 months | Cu, Pb, Cd, and Zn | Increasing the soil pH, and decreasing the acid-soluble fraction of Pb 51% and 16% of Zn at a 6% application rate of biochar. | [49] |
| Bamboo and paulownia biochar produced at 700–800 °C | BB: 10.0 PB: 10.5 | 2%, 4%, and 6% | 2 months | Cu, Pb, Cd, and Zn | Increasing the soil pH, and decreasing the acid-soluble fraction of Pb and Zn up to 22.12% and 24.31%, respectively, at a 6% biochar application rate. | [15] |
| Rice straw biochar produced at 350–500 °C | BC: 11.44 | 1%, 3%, and 5% | 30, 60, and 90 days | Pb, Zn, and Cd | Increasing soil pH, and decreasing exchangeable fraction of Pb and Zn up to 57% and 47.5%, respectively, after 30 days at 5% of biochar application rate. | [20] |
| Peanut shell biochar produced at 400 °C and 600 °C | PB400: 10.9 PB400: 11.3 | 3%, 5%, and 10% | 1 month | Zn, Pb | Biochar could increase the pH of the soil and respectively decrease 44.44% and 26.6% of the exchangeable fraction of Pb and Zn at the 10% biochar application ratio. | [23] |

### 3.5. Mechanism for Immobilizing Heavy Metals in Incubated Soil

The mechanism of heavy metal immobilization occurring in contaminated soil after incubation with biochar is still unclear since it is a complicated process with various means such as complexation, reduction, cation exchange, electrostatic attraction, and precipitation reaction, which might take place coincidently [17]. Metal adsorption on the biochar's surface might result in the immobilization of metals. There are two types of cation adsorption on organic materials and clay minerals: non-specific and specific adsorption [51]. Metals can bind to one another in the diffuse electric double layer by a process known as non-specific adsorption, which is governed by Coulombic interactions alone. The adsorption of metals in the inner layer that forms coordination bonds to the surface is referred to as "specific adsorption" [51,52]. Pb and Zn specifically interact with organic matter, while most

alkali and alkaline earth cations are adsorbed non-specifically [53]. As a result, biochar may immobilize metals through selective and non-specific adsorption [51]. In general, biochar eliminates HMs primarily through adsorption and immobilization [54]. Heavy metal remediation especially integrates both physical (physisorption and electrostatic attraction) and chemical reactions (precipitation, ion exchange, the complexation of functional groups, and reducing forms) [55].

In the present research, the reduction in the exchangeable fractionations of Pb and Zn after incubation with biochar and apatite ore could be associated with the characteristics of biochar such as high pH value (9.5), functional groups (OH, C-H, C=O, and C-O-C), the mineral in the ash (Ca, K, Mg, and Na), and porous and large area surface (especially PSB300). In addition, apatite also had specific properties, including high pH (8.9), rich in inorganic elements such as P, Si, Mg, and O, and various functional groups like phosphate, carbonate, and hydroxy, which could combine with heavy metals via the precipitation, exchange, and complexation reactions. Previous studies have reported that the primary reason for the immobilization of toxic metals in polluted soil is the rise in soil pH and EC after the incubation with amendments. The reason for the change in soil pH was that after 4-week incubation with biochar and apatite, soluble ions such as cations (Ca, Mg, Al, Si, Fe, and Na) or anions (Cl, $SO_4$, $CO_3$, and $PO_4$), and organic substances could be released from the biochar and apatite's particles into the soil solution. Additionally, small mineral fragments, particularly, Fe/Mn/O, and biochar fragments, may disseminate owing to redox reactions [20]. Therefore, the pH and EC of the soils surrounding these particles would rise, and there may be interactions with the heavy metals as a result [24].

Furthermore, the formation of $Pb_3(CO_3)_2(OH)_2$ or $Zn(OH)HCO_3$ due to the cation exchange and precipitation interactions between basic elements ($CO_3^{2-}$, $OH^-$, and other alkaline earth $Ca^{2+}$ or $Mg^{2+}$) or between biochar and Pb(II) or Zn(II) can significantly immobilize Pb and Zn. In this study, all amendments including PSB300, CCB300, and AP had high pH and EC values, which enabled the exchange reaction and hydroxy precipitation with zinc and lead ions in the soil solution. Thus, exchange and precipitation reactions may significantly contribute to immobilizing heavy metals in contaminated soil [24]. In addition, apatite, which is rich in phosphate, could form $Pb_5(PO_4)_3Cl$ and $Pb_5(PO_4)_3OH$ after incubation with contaminated soil. These substances are very stable and their formation might contribute to the increase in the F5 fractions of Pb and Zn. This process made heavy metals in the soil immobile by converting them from their mobile form into more stable forms, illustrating the utility of employing apatite and biochar to mitigate lead and zinc in contaminated soil. This reason can explain why even though apatite has a large surface area like biochar, it still affects the reduction in the heavy metal exchangeable fractions since it has numerous elements, functional groups, and a high pH value. In general, the effects of PSB300 on the decrease in F1-Pb and F1-Zn in incubated soil were significantly better compared to CCB300 at the same application ratios. This may be because of the much larger surface area of PSB300 compared to CCB300. Moreover, the biochar developed an organo-mineral layer on its surface, in addition to the formation of insoluble stable compounds such as $Pb_5(PO_4)_3Cl$ and $Pb_5(PO_4)_3OH$, after 4-week incubation with amendments, which could be used to explain the formation of stable fractions. This layer was made up of micro-agglomerates of nanosized minerals and inorganic compounds bound together by organic chemicals, some of which were humic substances [20,56]. These micro-agglomerates produced micro-aggregates when the biochar was separated, which may be incorporated into the stable micro-aggregate portion of the soil [20].

In numerous previous studies, FT-IR and EDS have been applied to test the biochar before and after incubation in contaminated soil to ascertain the dominant mechanism [57–61]. The present study's IR results showed changes in the IR spectra of biochar after incubation compared to before. The peak position and the peak intensity changed noticeably (Figure 5). The peak intensity at about 3442, 2922, 2370, and 1602 cm$^{-1}$ reduced significantly after incubation compared to before incubation, indicating that the functional groups of OH, C-H, C≡C, and C=O might be oxidized or combined with heavy metal in the soil solution.

Moreover, the peak at 796 cm$^{-1}$ indicated the adsorption of heavy metals with biochar [62] and the peak at around 464–669 cm$^{-1}$ confirmed the combination of phosphate with heavy metals [60].

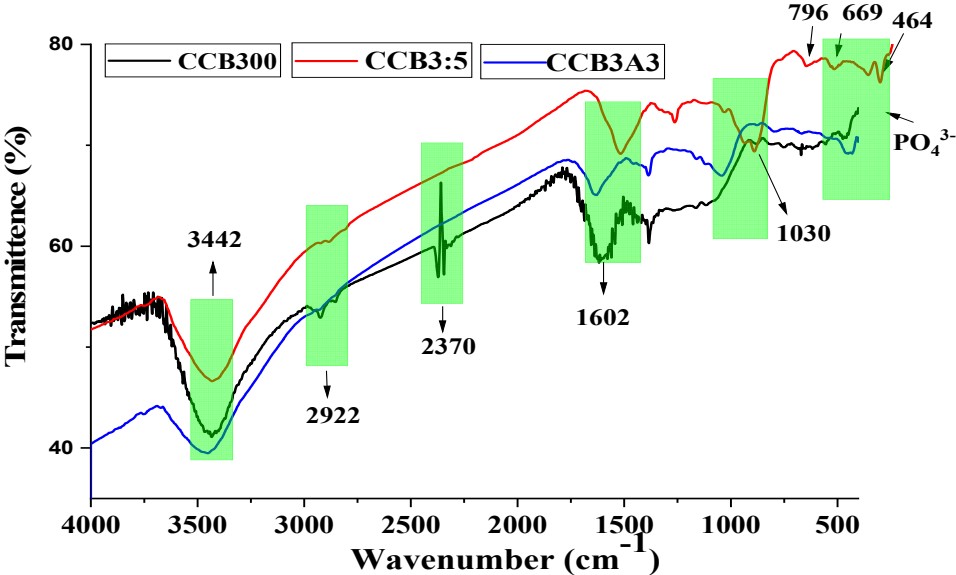

**Figure 5.** FT-IR of CCB300 before (CCB300) and after incubation with contaminated soil (CCB3:5 and CCB3A3).

The EDS results showed a considerable difference in elements on the biochar surface before and after incubation in contaminated soil (Figure 6). It was manifested that there were more Pb and Zn presenting on the surface of biochar after incubation (Figure 6B,C,E,F) compared to before incubation (Figure 6A,D), indicating that Pb and Zn combined with functional groups on the biochar surface, or accumulated on the surface by precipitation or physical adsorption.

To sum up, it was speculated that the dominant mechanism of the immobilization of heavy metals after one-month incubation with biochar and apatite in the present study included diverse processes such as precipitation, exchange, complexation reactions, and adsorption, which co-occurred in the soil solution.

*3.6. Correlation of the Exchangeable Fraction of Lead and Zinc with pH, OC, and EC of Incubated Soil after a 30-Day Incubation*

The exchangeable fraction is the most mobile in all chemical fractions of heavy metals and poses a severe environmental threat. The most crucial purpose of the heavy metal remediation of biochar is to reduce this fraction. Thus, the Spearman correlation was used to determine the relationship between the exchangeable fractions of Pb (F1-Pb) and Zn (F1-Zn) and the fundamental soil characteristics, including pH, OC, and EC. The correlation results of these factors are illustrated in Figure 5.

- Correlation of F1-Pb with pH, OC, and EC

Figure 7A illustrates the correlation results of F1-Pb with pH, OC, and EC and the results showed that F1-Pb had a moderate negative linear relationship with pH (r = −0.49) and EC (r = −0.49), while it had a strong negative linear relationship with OC (r = −0.72). This means that the more the pH, OC, and EC values rose, the less F1-Pb was. Hence, biochar and apatite had high pH, OC, and EC values that could facilitate immobilized lead in contaminated soil. In the meantime, pH, had a strong positive linear relationship with OC (r = +0.76) and EC (r = +0.80), whereas OC and EC had an exclusively strong positive correlation with r = +0.94. The negative correlation of F1-Pb with OC and pH was

reported in previous studies after the incubation of rice straw biochar or biochar/apatite in contaminated soil [19,20].

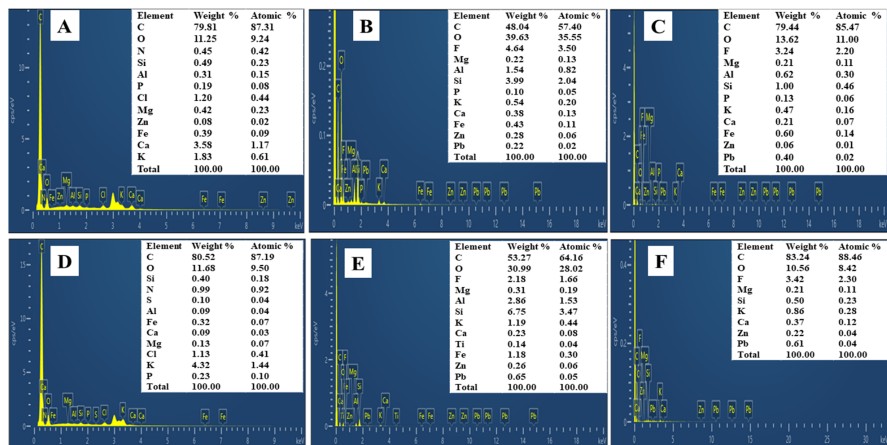

**Figure 6.** EDS results of Biochar before and after being incubated with heavy metal-contaminated soil: (**A**) EDS of PSB300 before incubation; (**B**) EDS of PPB3:5 (incubated PSB300 with soil sample with a ratio of 5% weight); (**C**) EDS of PSB3A3 (incubated PSB300 with apatite at the ratio of 3:3%); (**D**) EDS of CCB300 before incubation; (**E**) EDS of CCB3:5 (incubated CCB300 with soil sample with a ratio of 5% weight); (**F**) EDS of CCB3A3 (incubated CCB300 with apatite at the ratio of 3:3%).

- Correlation of F1-Zn with pH, OC, and EC

Figure 7B indicates that F1-Zb had a strong negative association with pH (r = −0.80) and OC (r = −0.78), while the correlation of F1-Zn with EC was a moderate negative linear relationship (r = −0.67). Meanwhile, the results illustrated that the relationship between pH, OC, and EC was very strong and positive, with r = +0.76 and 0.80, respectively, while OC and EC had a very strong positive linear relationship with r = +0.94. This result was consistent with previous studies [19,20,24,25], which showed that the soil pH and OC had a strong negative correlation with F1-Zn after incubating various biochars in contaminated soil.

Overall, higher amendment application rates led to more significant increases in pH, OC, and EC values while decreasing the exchangeable fractions of heavy metals. These factors could significantly influence the immobilization of heavy metals in polluted soil. Biochar had high pH, EC, and OC values, while apatite ore had high pH and EC values. As a result, these amendments could significantly alter these factors in contaminated soil after one-month incubation.

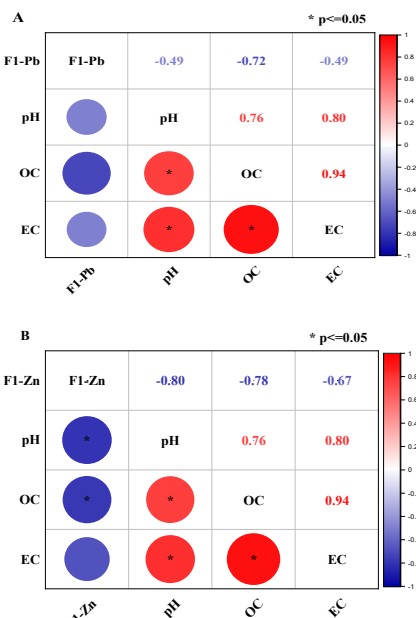

**Figure 7.** Correlation of pH, OC, and EC with the exchangeable fraction of Pb (**A**) and Zn (**B**).

## 4. Conclusions

(i) The properties such as pH, OC, EC, and surface morphology of corn cob and peanut-shell-derived biochar pyrolyzed at 300 °C (PB300 and CB300) were analyzed using various techniques, namely, SEM, EDX, FT-IR, and BET.

(ii) The changes in soil properties, including pH, OC, and EC, were investigated after 30-day incubation with biochar and apatite at 3% and 5% application rates, respectively. The results indicated that the studied soil's pH, OC, and EC values significantly increased with the application ratio of the amendment.

(iii) The impacts of biochar and the blend of biochar–apatite on the chemical speciation of Pb and Zn, especially the exchangeable fraction, in the studied field soil, were studied using Tessier's sequential extraction procedure. The results showed that the amendment could considerably decrease the exchangeable fractionations of Pb and Zn, which are mobile and detrimental to the ecosystem. The decline in the Pb and Zn exchangeable fractions was highest at 26% and 33%, respectively, compared to the control soil after the incubation. Meanwhile, the amendments could increase the Pb and Zn content in the organic and residual fractions by transforming them from the unstable fraction (F1) into stable fractions (F4 and F5), resulting in immobility in natural conditions.

(iv) CCB300 had a slightly better impact on the reduction in F1-Pb and F1-Zn than that of PSB300. This finding could be attributed to the difference in pH, OC, and EC values of PSB300 and CCB300 when CCB300 had higher pH, OC, and EC values than PSB300, though PSB300 had a larger surface area than CCB300.

(v) This study was conducted in a short time in laboratory conditions. Further research is required to determine how these amendments will affect the mobility and bioavailability of Pb and Zn in multi-contaminated soil in field conditions in a long-term experiment.

**Supplementary Materials:** The following supporting information can be downloaded at: https://www.mdpi.com/article/10.3390/su151511992/s1, Table S1: Operating parameter of the microwave digestion system for digesting soil samples for analyzing heavy metals; Table S2: ICP-MS Agilent 7900 parameters and the recovery values of heavy metals; Table S3: Tessier's sequential extraction process; Table S4: The percentages of Pb and Zn's chemical fractionations in the soils following one-month incubation with biochar (PSB300, CCB300) and the mixture of biochar/apatite.

**Author Contributions:** Conceptualization, T.X.V.; Methodology, T.X.V., T.T.T.N. and T.T.H.P.; Software, T.T.T.N. and T.T.H.P.; Validation, T.X.V. and T.T.T.N.; Data curation, T.X.V.; Writing—original

draft preparation, T.X.V.; Writing—review and editing, D.T.N.P.: Visualization, T.T.H.P. and D.T.N.P. All authors have read and agreed to the published version of the manuscript.

**Funding:** This research received funds sponsored by the Vietnamese Ministry of Education and Training under project B2020-TNA-15.

**Institutional Review Board Statement:** Not applicable.

**Informed Consent Statement:** Not applicable.

**Data Availability Statement:** Not applicable.

**Acknowledgments:** The authors acknowledge the Vietnamese Ministry of Education and Training under project B2020-TNA-15.

**Conflicts of Interest:** The authors declare no conflict of interest.

**Sample Availability:** Samples of the compounds CCB300 and PSB300 are available from the author.

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
