# Peer review of "Effects of Biochar and Apatite on Chemical Forms of Lead and Zinc in Multi-Metal-Contaminated Soil after Incubation: A Comparison of Peanut Shell and Corn Cob Biochar"

_sustainability, doi:10.3390/su151511992_

Round 1
Reviewer 1 Report (Previous Reviewer 4)
Dear Sir,
Greetings!
Based the correction made by the author and work the paper can be accepted for publication i don't find any thing that need to be changed.
BR
Author Response
Thank you for your comments. Please see the attachment.

Reviewer 2 Report (New Reviewer)

The Abstract, Discussion, and Conclusions are very difficult to read in some sections of the article. The manuscript needs thorough proofreading by a professional English speaker.
Author Response
Thank you for your comments. Please see the attachment.

Reviewer 3 Report (New Reviewer)
Comments
In this manuscript, the authors showed effects of the incubation of biochar and apatite on lead and zinc’s chemical fractions in multi-contaminated soil. The title accurately reflects the study. The objective is well defined and I have no criticisms regarding the interpretation of results. However, I think that the article is not ready for publication as it stands. The questions are as follows:
1. Before proceeding to describe your experiment, materials and actions, please describe your scientific hypothesis, concepts and the relevant reasoning for choosing the particular modelling approach. This should be accompanied by an overall description of the followed procedure.
2. Normally you do not put abbreviations in an abstract.
3. No mention of hypothesis you were testing or measurable aims of the research.
4. Need aim with measurable targets to determine success or not as the case may be.
5. No research questions to support hypothesis?
6. Study lacks depth and scientific rigor.
7. In the introduction, you need to connect the state of the art to mesocosm evaluation of the safety in the use of reclaimed water regarding emerging pollutants. Currently, this is not performed in a convincing way. Please follow the literature review by a clear and concise state of the art analysis. Please reason both the novelty and the relevance of your paper goals.
8. The novelty of this work is fair. However, a quick search reveals that this study does not differ significantly from other publications that were published by the authors. The authors are asked to show their original contribution in this field in a more convincing way.
9. If this is an experimental study, I think experimental data should be shown in the corresponding Figure as well as fitting model. How many runs does the experimental procedure consist of ?
10. Please improve the clarity of figure 1.
11. Please add a statistical explanation to the original text
12. Please review references format.
Minor editing of English language required.
Round 2
Reviewer 2 Report (New Reviewer)
The authors have taken into account the aforementioned criticisms. It is advised to publish this article.
Reviewer 3 Report (New Reviewer)
All comments have been addressed properly, so the article suggested to publish with its present form
This manuscript is a resubmission of an earlier submission. The following is a list of the peer review reports and author responses from that submission.
Round 1
Reviewer 1 Report
1. low novelty. You can easily find numerous studies on the same topic.
2. experimental design too simple. Just a incubation study and a sequential extraction of Zn and Pb is not enough for publication.
English is fine
Reviewer 2 Report
Major concerns:
-Some previous studies have reported positive results of combining biochar with apatite ore in heavy metal remediation. Then, the novelty of this study should be strengthened in the Instruction section.
-Although the exchangeable fraction of Pb and Zn was decreased after one-month amendment incubation, various forms of heavy metal residues could not be truly removed. Therefore, the disadvantage of this method should be discussed, and the future study direction in this field should be indicated.
-The metal absorption of biochar and apatite lacks selectivity. The absorption of this material toward various metallic elements, such as Fe(II), Ca(II), Mg(II), Cu(II), and so on, should be mentioned and discussed.
Minor comments:
Line 342: “the F5- Pb contributed almost 50% of the total,” Is this description right? F2?
Line 408: 663.4 ± 16.2 mg kg-1? It should be checked.
Line 538: “mitigate and zinc”, lead is missing.
Line 586: “induce” should be reduce?
Author Response
Please see the attachment. We sicerely apologize for the inconvenience. Please ignore the previous file.

Reviewer 3 Report
The paper " Immobilizing Lead And Zinc In Multi-Contaminated Soil Using Biochar And Apatite: A Comparison Of Peanut Shell And Corn Cob Biochar " treat a simple topic with several parameters. However few questions are reported in the attachment.

The english language is fine, however several phrases need a reorganization.
Reviewer 4 Report
Thanks for the invitation to review the paper “Immobilizing Lead And Zinc In Multi-Contaminated Soil Using
Biochar And Apatite: A Comparison Of Peanut Shell And Corn Cob Biochar”
The following are the suggestion
## Need to be modify Besides, Pb and Zn in the studied soil were distributed in the sequence of F2
> F5 > F1 > F3 > F4 and F5 > F2~F3 > F1 > F4, respectively. After 30-day incubation, Pb and Zn’s
exchangeable fractions in incubated soil were significantly diminished ( p < 0.05) up to about 28%
compared to the control soil. Meanwhile, the incubation of biochar and apatite could significantly
increase the Pb and Zn’s organic (F4) and residual fractions (F5) ( p < 0.05), especially at the application ratio of 5% biochar and 3:3% biochar/AP.
##Avoide title keywords Heavy metal immobilization; peanut shell biochar; corn cob biochar; heavy metal speciation; heavy metal contamination;
## Need to revised and it should be linked with Lead And Zinc and biochar
Heavy metal pollution has been a severe issue worldwide. A serious environmental |
problem is the accumulation of heavy metals in the soil brought on by industrial and hu- |
man activities [1,2] . The main reasons caused heavy metals pollution in soil were at- |
tributed to various factors, such as mining and smelting operations or the usage of pesti- |
cides and sewage sludge [3–5]. Heavy metals can accumulate and remain in the soil for |
an extended period since they are not biodegradable [5]. Besides, they can be mobile in |
soil depending on their chemical state in soil and different physicochemical processes [6]. |
As a result, they can intrude into the food chain and pose a severe threat to human health |
due to detrimental impacts [4,7,8,9]. |
## pH and EC is measure in kcl
The pH and EC values of the soil, biochar and apatite were determined using |
a Hanna HI 9124 pH meter (Rumani). The materials were combined in a 1:10 (w/v) ratio |
with distilled water, stirred, and allowed to stand for one hour before measuring [26]. The |
pipette method was used to analyze the tested soil's soil texture [25,26]. The C/N multi |
3100 (Analytik Jena, Germany) was utilzed to analyze the material's organic carbon (OC) |
content in biochar and apatite) [27]. |
##check the pH value
6.69 ± 0.02 |
9.53 ± 0.01 |
9.50 ± 0.01 |
8.99 ± 0.01 |
## Conclusions section should be revised
rest paper is written well.
